# Dietary Patterns during Pregnancy Are Associated with the Risk of Gestational Diabetes Mellitus: Evidence from a Chinese Prospective Birth Cohort Study

**DOI:** 10.3390/nu11020405

**Published:** 2019-02-15

**Authors:** Jiajin Hu, Emily Oken, Izzuddin M. Aris, Pi-I D. Lin, Yanan Ma, Ning Ding, Ming Gao, Xiaotong Wei, Deliang Wen

**Affiliations:** 1Department of Social Medicine, School of Public Health, China Medical University, Shenyang 110122, China; jjhu@cmu.edu.cn (J.H.); cmugaom@163.com (M.G.); xtwei@cmu.edu.cn (X.W.); 2Division of Chronic Disease Research Across the Lifecourse, Department of Population Medicine, Harvard Medical School, Boston, MA 02215, USA; emily_oken@harvardpilgrim.org (E.O.); Izzuddin_Aris@harvardpilgrim.org (I.M.A.); pil864@mail.harvard.edu (P.-I.D.L.); 3Department of Nutrition, Harvard T.H. Chan School of Public Health, Boston, MA 02113, USA; 4Department of Obstetrics and Gynecology, Yong Loo Lin School of Medicine, National University of Singapore, Singapore 119228, Singapore; 5Agency for Science, Technology and Research, Singapore Institute for Clinical Sciences, Singapore 119228, Singapore; 6Department of Environmental Health, Harvard T.H. Chan School of Public Health, Boston, MA 02113, USA; 7Research Center for Environmental Medicine, Kaohsiung Medical University, Kaohsiung 80709, Taiwan; 8Department of epidemiology and health statistics, School of Public Health, China Medical University, Shenyang 110122, China; ynma@cmu.edu.cn; 9Department of Curriculum and Teaching Research, Research Center of Medical Education, China Medical University, Shenyang 110122, China; nding@cmu.edu.cn

**Keywords:** pregnancy, maternal nutrition, dietary pattern, gestational diabetes mellitus, Asia, three-day food diary, food frequency questionnaire

## Abstract

Dietary patterns during pregnancy have been shown to influence the development of gestational diabetes mellitus (GDM). However, evidence from Asian populations is limited and inconsistent. We conducted a prospective cohort study in China to assess the relationship between dietary patterns and GDM. We administered three-day food diaries (TFD) and food frequency questionnaires (FFQ) at the second trimester. GDM was diagnosed with a 75 g 2-h oral glucose tolerance test at 24–28 weeks of gestation. We identified dietary patterns using principal components analysis and used multivariable logistic regression to investigate associations of dietary patterns with GDM. Of the 1014 participants, 23.5% were diagnosed with GDM. Both the TFD and FFQ identified a “traditional pattern”, consisting of high vegetable, fruit, and rice intake, which was associated with a lower GDM risk (odds ratio (OR) for quartile 4 versus quartile 1: 0.40, 95% CI: 0.23–0.71 for traditional pattern (TFD); OR: 0.44, CI: 0.27–0.70 for traditional pattern (FFQ)). The protective associations were more pronounced among women ≥35 years old. A whole grain–seafood TFD pattern was associated with higher risk of GDM (OR: 1.73, 95% CI: 1.10–2.74). These findings may provide evidence for making dietary guidelines among pregnant women in Chinese populations to prevent GDM.

## 1. Introduction

Gestational diabetes mellitus (GDM), which is the onset of diabetes or glucose intolerance first recognized during pregnancy [1], is a common medical disorder with short- and long-term adverse outcomes for both mothers and children [2,3,4,5]. Generally, Asian populations are at a higher risk of GDM compared to the western population [6,7]. In China, there was a rapid increase in the prevalence of GDM during the last decades, and the reported prevalence currently ranges from 9.3% to 18.9%, depending on the region [8,9].

Increasing evidence suggests that dietary intake has an important role in the development of GDM [10,11,12,13,14,15,16,17,18]. Most studies have examined the role of individual macronutrients or micronutrients on the subsequent risk of GDM [19,20,21,22]. However, the “single nutrient” approach may be inadequate, considering the complicated interactions among foods and nutrients. Instead of looking at individual nutrients or foods, examining the effects of overall dietary patterns, which represent a broader picture of food and nutrient consumption, may be more relevant and useful for predicting future disease risk [23]. 

Among previous studies that examined maternal dietary patterns in relation to GDM risk [12,13,14,16,20,24,25,26,27,28,29,30,31,32,33], few studies were based in Asia [24,25,26,27,28], where there is a rapid transition from a traditional diet to a more westernized diet [34]. Furthermore, these studies [24,25,26,27,28] have reported conflicting results. For example, in a large birth cohort in southern China [24], a vegetable-based diet was observed to be linked with a lower risk of GDM, while another study [28] found no associations between plant-based dietary patterns and GDM risk. A study in Singapore [27] reported that a seafood–noodle pattern was associated with a lower GDM risk, while a study in China [24] reported a positive association between a seafood–sweet dietary pattern and GDM. 

Another limitation from previous studies is that a single dietary assessment tool was used, such as food frequency questionnaire (FFQ) [24,26] or food diaries/dietary recalls [12,27]. FFQ and food diary/dietary recall methods both carry inherent limitations in measuring food consumptions [35]. While FFQs have the advantage of capturing long-term habitual diet, they have limitations in accurately and prospectively recording food intake, and often overestimate consumption [35,36,37]. Food diaries/dietary recalls provide more accurate assessment of food intake, but fail to reflect stable and long-term food intake [35]. Using both methods to assess dietary patterns will give a more comprehensive capture of diet and thus lead to a more robust conclusion.

The objective of the present study was to use an FFQ and three-day food diaries (TFD) to assess dietary patterns among pregnant women in a Chinese prospective birth cohort study, and to examine the association of dietary patterns during pregnancy with the risk of GDM.

## 2. Materials and Methods 

### 2.1. Study Population

We established a multicenter prospective birth cohort, “Born in Shenyang Cohort Study” (BISCS), to examine associations of prenatal factors in relation to maternal and child health. Between April and September 2017, we enrolled pregnant women at 21–24 weeks of gestation from 54 hospitals and community health care centers, which were all the perinatal care institutes provide antenatal and maternity care in the urban areas of Shenyang, located in northeast China. The number of participants each institute recruited was calculated by using the amount of primary perinatal care of the institute in the last year divided by 10. We included women with single pregnancies at 21–24 weeks of gestation, without any mental disease, and who intended to remain in Shenyang for the next three years. After screening, a total of 2068 women were eligible to participate in the study, and 1338 agreed to participate. All participants provided written informed consent, and the ethics committee of China Medical University approved the study.

We conducted face-to-face interviews at the enrollment visit using structured questionnaires (*n* = 1338; gestational age (mean ± SD): 22 ± 1.2 weeks) to obtain information on demographic and socio-economic status, personal lifestyle, environmental exposures, and medical history. Participants were followed up at 24 (±1.2) weeks of gestation for oral glucose tolerance testing (OGTT). For the present analysis, we excluded those who reported previous diabetes (type 1 or type 2; *n* = 3), had missing or incomplete records on OGTT (*n* = 165), had an incomplete TFD or FFQ (*n* = 40), or implausible values of total energy intake (< 500 or >3500 kcal/day) [38] (*n* = 116). A total of 1014 women were included in the final analysis. 

### 2.2. Exposure: Prenatal Diet

We used both three-day food diaries and food frequency questionnaires to assess dietary intake before OGTT.

At the enrollment visit, clinical research staff trained participants to correctly complete the TFD. A visual-aid booklet with colored photographs of 200 common food items with different portion sizes was provided to each participant. Participants were instructed to write down all food and drink consumed, including the list of all ingredients and their portions (in grams), over two consecutive weekdays and one weekend day; participants filled out the TFD before OGTT at 24 (±1.2) weeks of gestation. We collected the completed TFD when participants returned for their OGTT. The handwritten TFD was entered into an electronic database, and we summarized food items into 21 non-overlapping food groups (Appendix A). We calculated daily food intake values by averaging the consumption level for each food group over three days, and derived daily energy and nutrient intakes according to China Food Composition Database [39]. 

We also used a semi-quantitative FFQ to assess dietary intake throughout early pregnancy at the enrollment visit (mean gestation weeks ± SD: 22 ± 1.2 week). The FFQ consisted of 25 food items and beverages, including nine frequency categories ranging from “almost never eat” to “three times or more per day”. Participants reported their food intake frequency over the past two months. Data on the amount consumed were not collected in the FFQ, thus we calculated only the daily consumption frequency for each food group. To be comparable with the TFD, we further combined the 25 FFQ food groups into the 21 TFD food groups (Appendix A) based on their similarity in nutrient profiles (e.g., combined pork, beef, and lamb as a red meat group). We evaluated the reproducibility of the FFQ after combined (21 food groups), based on a subset of 401 non-GDM participants who completed a second FFQ at 31–35 gestational weeks (mean gestation weeks ± SD: 33 ± 1.1). We chose non-GDM women because GDM participants may change their diet habits after the OGTT in response to dietary advice. Crude Spearman correlation coefficients of consumption frequencies (per day) captured by the two FFQs ranged from 0.43 (tubers) to 0.80 (vegetables). We also compared the consumption frequency (per day) between FFQ and TFD for each food group, and the Spearman correlation coefficients ranged from 0.25 (whole grains) to 0.60 (vegetables). 

### 2.3. Outcome: Gestational Diabetes Mellitus

At 24–28 weeks of gestation, participants took a 75 g, 2-h oral glucose tolerance test. All participants had overnight fasting of at least eight hours before OGTT. We defined GDM using criteria recommended by the International Association of Diabetes Pregnancy Study Group: fasting plasma glucose ≥ 5.1 mmol/L, 1-h plasma glucose ≥ 10.0 mmol/L, or 2-h plasma glucose ≥ 8.5 mmol/L [40].

### 2.4. Covariates

We collected information on participants’ age, ethnicity, educational attainment, household income, smoking status, parity, history of diabetes, physical activity, and self-reported pre-pregnancy weight at recruitment. We measured participants’ height using a calibrated stadiometer at recruitment. Age (in years) was treated as a continuous variable, except for the descriptive statistic for which we categorize into four groups (<25, 25–29, 30–34, ≥35 years). Ethnicity was categorized into two groups (Han versus Others). Education attainment was classified into four groups (middle school or below, high school, college, graduate or above). Annual household income was reported in China Yuan (1 China Yuan = 0.14 US Dollar) and was categorized into four categories (<¥30,000, ¥30,000–< ¥50,000, ¥50,000–< ¥70,000, ≥¥70,000). Pre-pregnancy body mass index (BMI) was calculated as pre-pregnancy weight (kg) divided by height (m) squared, and grouped into four categories (<18.5, 18.5–< 23.0, 23.0–< 25.0, ≥25.0 kg/m^2^) using WHO references for Asian populations [41]. Smoking status during pregnancy was treated as a dichotomized variable (yes/no). Parity was divided into two categories (0, ≥1). Physical activity during pregnancy was measured with a Chinese version of the Pregnancy Physical Activity Questionnaire (PPAQ), a validated tool for the measurement of physical activity in Chinese pregnant women [42]. We used multiple imputation to impute missing values (*n* = 38) for household income. No missing value was observed for all other covariates.

### 2.5. Statistical Analyses

To identify dietary patterns, we conducted factor analysis (principal component analysis) for TFD and FFQ. Varimax rotation was applied for greater interpretability. We standardized the daily food intake calculated from TFD and daily consumption frequency calculated from FFQ, and used the standardized values of the 21 food groups to derive dietary patterns. We identify distinct dietary patterns based on eigenvalue, factor interpretability after varimax rotation, and a scree plot (Appendix A), showing the proportion of the variance of total consumption of the food variables. For the TFD, the dietary pattern scores were calculated by summing the standardized food intake weighted by corresponding factor loadings. For the FFQ, the dietary pattern scores were calculated by summing the standardized food consumption frequencies weighted by corresponding factor loadings. Food items with a factor loading greater than |0.2| [27] were defined as the main contributors to the dietary patterns. We categorized participants into quartiles based on their dietary pattern scores for subsequent analyses. 

We used chi-square tests to compare characteristics between GDM and non-GDM participants. We used *t*-tests or ANOVA tests to compare pattern scores across different social demographic variables. We used logistic regression models to estimate the odds ratio (OR) and 95% confidence interval (CI) for GDM in relation to dietary pattern quartiles, using the lowest quartile of the dietary pattern score as reference. To test for a linear trend, we reported *P*-for-trend by taking the median score of each quartile of dietary patterns and analyzing it as a continuous variable in multivariable models. We used multivariable linear regression models to estimate associations between dietary pattern scores and glucose levels. We conducted crude and adjusted analyses using the following models: Model 1, the crude model (individual dietary pattern); Model 2, which is Model 1 plus other dietary patterns derived from the same dietary assessment tool (TFD or FFQ); and Model 3, which is Model 2 plus pre-pregnancy BMI, age, parity, family income, education level, ethnicity, smoking status, total energy intake, and physical activity. As a subsidiary analysis, we also adjusted for dietary iron intake, which might mediate associations of dietary patterns with GDM, as well as iron supplements during pregnancy. Previous studies have shown that associations between dietary patterns and GDM may vary by maternal characteristics [24,26,28]; therefore, we examined potential effect modification by age (<35 years versus ≥ 35 years), pre-pregnancy weight status (BMI < 23 kg/m^2^ versus BMI ≥ 23 kg/m^2^), and parity (0 versus ≥1) by including multiplicative interaction terms in the models. We further used logistic and linear regression models to assess associations of intake of the 21 food groups with risk of GDM and with plasma glucose levels, after adjusting for maternal characteristics and all other food items.

We performed all analyses using Stata S.E. version 13 (Stata Corp, Texas, TX, United States).

## 3. Results

### 3.1. Characteristics of Participants

The incidence of GDM was 23.5% in our study population (238 out of 1014 pregnant women). There were no significant differences for age, ethnicity, and parity between the women who participated in the cohort study (*n* = 1338) and women did not respond (*n* = 730). Women who participated in the present study reported a higher rate of educational attainment of college and above (75.2% versus 69.7%). There were no significant differences for educational attainment, household income per year, smoking status, and physical activity level between GDM and non-GDM women (Table 1). However, compared to women without GDM, women with GDM tended to be older (≥35 years old at recruitment), of minority (non-Han Chinese) ethnicity (30.3% versus 19.6%), were more likely to have a BMI higher than 25.0 kg/m^2^ before pregnancy (31.5% versus 15.6%), and had energy intake of less than 2100 kcal/day (67.7% versus 59.0%) (*p* < 0.05 for all).

### 3.2. Dietary Patterns

For the TFD, we identified four dietary patterns accounting for 26.4% of the total variation (Table 2). The first pattern, which we named the “Traditional pattern (TFD)”, was characterized by a high intake of tubers, vegetables, fruits, rice, red meat, eggs, and nuts. The second pattern, which we named “Sweet foods pattern (TFD)”, was characterized by high intake of pastry and candy, sweet beverages, shrimps, crabs, mussels, fruits, and red meat. The third pattern, which we named the “Fried food–beans pattern (TFD)”, was characterized by high intake of fried foods, beans and products, and dairy products, and a low intake of organ meats. The fourth pattern, which we named “Whole grain-seafood pattern (TFD)”, was characterized by a high intake of whole grains, shrimps, crabs, mussels, nuts, and seaweed, and a low intake of eggs, dairy products, and rice. 

We also identified three dietary patterns from the FFQ (Table 2), which accounted for 32.4% of the total variation. The first pattern, which we named “Fish–seafood pattern (FFQ)”, was characterized by a high intake of marine fish, shrimps, crabs, mussels, freshwater fish, seaweed, and organ meat. The second pattern, which we named “Protein–sweets pattern (FFQ)”, was characterized by high intake of dairy products, milk, eggs, beans and products, nuts, pastries, and candies. The third pattern, which we named “Traditional pattern (FFQ)”, was characterized by a high intake of vegetables, fruits, rice, and nuts. The two traditional patterns derived from TFD and FFQ shared similar food composition and had significant correlation in pattern scores (Spearman correlation coefficient: 0.30, *p* < 0.001).

Table 3 presents dietary pattern scores in relation to participants’ characteristics. Women who had higher “Traditional pattern (TFD)” scores were younger, of lower educational attainment, and had higher daily energy intake. Women with higher “Sweet foods pattern (TFD)” or “Fried food–beans pattern (TFD)” scores had higher daily energy intake. Women who had higher scores for “Whole grain–seafood pattern (TFD)” were older and of higher educational attainment. Women who had higher “Fish–seafood pattern (FFQ)” scores were of higher educational attainment. Women with higher “Protein–sweets pattern (FFQ)” scores were more likely to be 25–29 years old, of higher educational attainment, and were more like to be primipara. Women with higher “Traditional pattern (TFD)” were more likely to be 25–29 years old, of higher educational attainment and higher household income level, had higher pre-pregnancy BMI, and higher physical activity level during pregnancy.

### 3.3. Dietary Patterns in Relation to Gestational Diabetes Mellitus

We observed lower odds of GDM in relation to higher traditional pattern quartiles (both TFD and FFQ), and higher odds of GDM in relation to a higher “Whole grain-seafood pattern (TFD)” quartile (Table 4). The adjusted ORs for the highest quartile compared with lowest among “Traditional pattern (TFD)”, “Traditional pattern (FFQ)”, and “Whole grain-seafood pattern (TFD)”, were 0.40 (95% CI: 0.23, 0.71; *p* for trend = 0.005), 0.44 (CI: 0.27, 0.70; *p* for trend < 0.001), and 1.73 (CI: 1.10, 2.74; *p* for trend = 0.007), respectively. No associations were observed for the “Sweet foods pattern (TFD)”, “Fried food–beans pattern (TFD)”, “Fish–seafood pattern (FFQ)” or “Protein-sweets pattern (FFQ)” in relation to GDM. Further adjusted iron supplements and dietary iron intake generally did not changed the results.

In the subgroup analyses (Figure 1), the protective effect of traditional patterns (both TFD and FFQ) was more pronounced among women of advanced age (≥35 years old). For the “Traditional pattern (TFD)”, compared with the lowest quartile, the adjusted OR of the highest quartile was 0.92 (95% CI: 0.33, 2.52) for women <35 years old, and 0.25 (95% CI: 0.12, 0.52) for women ≥35 years old. Similarly, compared with the lowest quartile of the “Traditional pattern (FFQ)” score, the OR of the highest quartile was 0.67 (95% CI: 0.31, 1.44) for women <35 years old, and 0.33 (95% CI: 0.18, 0.62) for women ≥35 years old. There were no modification effects for any dietary patterns by parity or pre-pregnancy BMI.

### 3.4. Dietary Patterns in Relation to Plasma Glucose Levels

In the adjusted model (Table 5), the “Traditional pattern (TFD)” score is inversely associated with fasting (blood glucose level (β): −0.03; 95% CI: −0.05, 0.00) and 1-h post-load plasma glucose (β: −0.11; 95% CI: −0.19, −0.03). The “Traditional pattern (FFQ)” score is inversely associated with fasting (β: −0.04; 95% CI: −0.06, −0.02), 1-h plasma glucose (β: −0.07; 95% CI: −0.13, −0.01), and 2-h plasma glucose (β: −0.07; 95% CI: −0.11, −0.02). The “Whole grain-seafood pattern (TFD)” score was positively associated with 1-h plasma glucose (β 0.11; 0.02, 0.19).

### 3.5. Food Groups Intake in Relation to Gestational Diabetes Mellitus Risk and Plasma Glucose Levels

We found certain food groups, including rice, fruits, and vegetables, to decrease the risk of GDM and glucose levels, while whole grain and dairy products were positively associated with glucose levels (Appendix A).

## 4. Discussion

In this Chinese prospective birth cohort study, we found common traditional dietary patterns derived from three-day food diaries (TFD) and food-frequency questionnaires (FFQ), which are comprised mainly of vegetables, fruits, and rice. Traditional patterns were significantly associated with a decreased risk of GDM and lower glucose levels. The associations of traditional patterns with the risk of GDM appears to be more pronounced in women ≥35 years old. In contrast, the “Whole grain–seafood pattern (TFD)” was positively associated with the risk of GDM and plasma glucose levels 1 hour after OGTT.

Our findings are in line with several previous studies [13,26,27,31]; for example, in a prospective cohort study in China [26], Zhou et al. found a high carbohydrate–low protein diet based on rice, wheat, and fruit was associated with a lower risk of GDM. In a multi-ethnic birth cohort study in Singapore [27], although a vegetable–fruit–rice-based diet was not associated with GDM in all subjects, it was associated with lower risk of GDM among 505 Chinese participants. Results from the Nurses’ Health Study II [13,31] also supported that dietary patterns based on vegetables and fruits are associated with a lower risk of GDM. Inconsistent with our findings, Du et al. [25] report that the traditional dietary pattern in China was positively associated with GDM risk, and Mak et al. [28] report that there is no association between a plant-based diet and risk of GDM among Chinese. This may be explained by the different combination of food groups; for example, in Du et al.’s [25] study, the traditional pattern did not include fruits, which have been shown to be protective against GDM among the Chinese population [43]. In our study, higher calorie intake was associated with lower risk of GDM. One of the possible reasons could be that in our study subjects, high energy intake (≥2100 kcal/day) was also associated with lower age, which is a protective factor against GDM. However, mechanisms are still needed to investigate. In our study, physical activity level was not associated with GDM; a possible reason for this could be that the Pregnancy Physical Activity Questionnaire (PPAQ) measured physical activity level over a short time (seven days) before OGTT, and thus may not represent the physical activity level during the whole early pregnancy.

The association between the vegetable–fruit–rice-based traditional dietary pattern and lower risk of GDM could be explained by several potential mechanisms. First, fruits and vegetables are rich in dietary fiber, which may reduce adiposity and improve insulin sensitivity [15,44]. Furthermore, dietary fiber intake could delay gastric emptying and slows food digestion and absorption, thus decreasing postprandial plasma glucose levels [15,45,46]. In addition, vegetables and fruits are rich in polyphenols and other antioxidant components such as vitamin C, vitamin E, and carotenoids [43,47]. These compounds may decrease the risk of GDM by mitigating the oxidative stress that interferes with glucose uptake in cells. In the present study, rice intake was associated with a lower risk of GDM as well as lower plasma glucose levels. A previous study in China also reported that diets characterized by a high intake of rice was associated with lower risk of GDM [26]. Although rice has a relatively high glycemic index [48], it also contributes to a high carbohydrate intake diet pattern, which was associated with lower risk of GDM [26,49]. Higher rice intake may also suggest a higher frequency of home-cooked meals, which are the traditional dietary practice in China, and thus less frequent exposure to the unhealthy high-energy-density food in restaurants, such as deep-fried foods. 

Another important finding was the protective effect of the traditional pattern diet on GDM risk that was more pronounced among women of advanced age (≥35 years old). Consistent with previous studies, we observed higher incidence of GDM among women ≥35 years old [50,51]. Women of older age are more likely to have lower insulin sensitivity [52]. Fibers from vegetables and fruits may improve their insulin sensitivity [15], thus decreasing plasma glucose levels and reducing the risk of GDM. This finding points out the importance of a healthy diet among women with advance reproductive age. With the introduction of the universal two-child policy in China, women’s childbearing age has risen rapidly [53], thus increasing the proportion of women who are at risk of developing GDM. Our finding could be helpful in setting dietary guidelines for women of advanced reproductive age to prevent GDM development.

In the present study, we also found that a “Whole grain–seafood (TFD)” dietary pattern was positively associated with a higher risk of GDM. However, the pattern was identified only from the TFD, but not the FFQ. A previous study had reported that diets rich in whole grains are associated with a lower risk of GDM [17]. The associations of seafood with GDM risk were inconsistent [24,27]. In our study, women who had a higher “Whole grain–seafood pattern (TFD)” score were older in age, and thus may have a higher risk of GDM. However, the odds ratio remained significant after adjusting for age (Model 3). Another mechanism that could explain the elevated risk of GDM is that whole grains and seafood might contain more arsenic and other environmental contaminants [54,55], which have been shown to be associated with increasing insulin resistance and impairing insulin production [56,57]. Further studies are still needed to investigate the mechanism.

A previous study [24] had found that a sweets and seafood pattern was associated with a higher risk of GDM. However, in our study, both “Sweet foods pattern (TFD)” and “Protein–sweets pattern (FFQ)” were not associated with a risk of GDM. This could be explained by that, in our study, pregnancies with higher “Sweet foods pattern (TFD)” and “Protein-sweets pattern (FFQ)” scores were younger in age and more likely to be uniparas, and thus may have a lower risk of GDM.

The strength of our study includes a population-based cohort design and a prospective assessment of dietary intake. Both the TFD and FFQ were conducted before the OGTT test, and thus it is unlikely that a diagnosis of GDM affected dietary reporting. Another strength is the joint use of two measurement methods to identify dietary patterns. The TFD provided an accurate assessment of short-term food intake, while the FFQ captured habitual diet across pregnancy. Finally, the present study focused on dietary patterns rather than individual nutrients or dietary components, which enabled us to investigate synergistic effects among various nutrients and foods. Findings from dietary patterns are also more relevant for clinical practice and public health intervention.

Our study has several limitations. First, the FFQ in the present study consisted of only 25 items, and thus was relatively brief. However, these 25 food items represented the most commonly consumed dishes in a Chinese population [39]. Comparison with TFD also showed a high proportion of overlap between these 25 food items and the food items recorded in the TFD. Second, our study was based in an urban setting with a relatively high socioeconomic status, this could partially explain the high incidence of GDM among our study subjects; however, according to a previous study, dietary patterns between rural and urban population are generally the same in China [58]. Third, we lacked measurements of dietary patterns at the first trimester or pre-pregnancy. However, previous studies showed that dietary patterns are likely to remain stable across pregnancy [59]. Finally, residual confounding, such as family history of diabetes, was not accounted for in this analysis. Pregnant women with a family history of diabetes may have higher risk of GDM, and are also more likely to choose a healthier dietary pattern during pregnancy, such as a traditional pattern. Thus, adjusting for family history of diabetes may further strengthen the inverse associations between traditional pattern scores and risk of GDM.

## 5. Conclusions

In conclusion, we found that a traditional dietary pattern based on vegetables, fruits, and rice was associated with lower risk of GDM and lower plasma glucose levels, and the effects were more pronounced among women ≥35 years old. In addition, a “Whole grain–seafood (TFD)” dietary pattern identified from TFD was associated with higher risk of GDM. These findings could be helpful in setting dietary guidelines to prevent GDM in a Chinese population.

## Figures and Tables

**Figure 1 nutrients-11-00405-f001:**
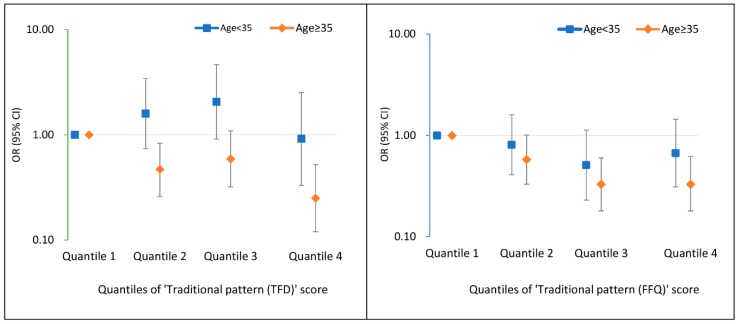
Associations between “Traditional pattern (TFD)” and “Traditional pattern (FFQ)” score quantiles and the risk of gestational diabetes mellitus, stratified by maternal age (<35 vs. ≥35). Adjusted for other dietary patterns derived from the same dietary assessment tool (TFD or FFQ), pre-pregnancy body mass index (BMI), age, parity, family income, education level, ethnicity, smoking status, total energy intake from three-day food diaries, and physical activity. TFD: three-day food diaries; FFQ: food frequency questionnaire.

**Table 1 nutrients-11-00405-t001:** Characteristics according to gestational diabetes mellitus status among 1014 pregnant women in the “Born in Shenyang” cohort.

Characteristics	All Participants, *n* (%)	Gestational Diabetes Mellitus, n (%)		
		No	Yes	*p*-value
	*n* = 1014	*n* = 776 (76.5)	*n* = 238 (23.5)	
Age at enrollment (Years)				<0.001
<25	61 (6.0)	53 (6.8)	8 (3.4)	
25–29	442 (43.6)	363 (46.8)	79 (33.2)	
30–34	357 (35.2)	263 (33.9)	94 (39.5)	
≥35	154 (15.2)	97 (12.5)	57 (24.0)	
Ethnicity				0.001
Han	790 (77.9)	624 (80.4)	166 (69.7)	
Minority	224 (22.1)	152 (19.6)	72 (30.3)	
Educational attainment				0.918
Middle school or below	86 (8.5)	64 (8.2)	22 (9.2)	
High school	147 (14.5)	111 (14.3)	36 (15.1)	
College	683 (67.4)	527 (67.9)	156 (65.6)	
Graduate or above	98 (9.6)	74 (7.3)	24 (10.1)	
Household income per year, CNY				0.678
<30,000	251 (24.8)	193 (24.9)	58 (24.4)	
30,000𠄓<50,000	266 (26.2)	197 (25.4)	69 (29.0)	
50,000𠄓<70,000	224 (22.1)	177 (22.8)	47 (19.7)	
≥70,000	235 (23.2)	178 (22.9)	57 (23.9)	
Missing Value	38 (3.8)	31 (4.0)	7 (2.9)	
Parity				0.001
0	790 (77.9)	624 (80.4)	166 (69.8)	
≥1	224 (22.1)	152 (19.6)	72 (30.2)	
Smoking before or during pregnancy				0.664
No	996 (98.2)	763 (98.3)	233 (97.9)	
Yes	18 (1.8)	13 (1.7)	5 (2.1)	
Pre-pregnancy BMI category, kg/m^2^				<0.001
<18.5	127 (12.5)	114 (14.7)	13 (5.5)	
18.5𠄓<23.0	540 (53.3)	429 (55.3)	111 (46.6)	
23.0𠄓<25.0	151 (14.9)	112 (14.4)	39 (16.4)	
≥25.0	196 (19.3)	121 (15.6)	75 (31.5)	
Physical activity, MET-hour/week				0.667
<100	264 (26.0)	200 (25.8)	64 (26.9)	
100 to <200	557 (54.9)	432 (55.7)	125 (52.5)	
≥200	193 (19.0)	144 (18.5)	49 (20.6)	
Energy intake, kcal/d ^a^				0.017
<2100	619 (61.1)	458 (59.0)	161 (67.7)	
≥2100	395 (38.9)	318 (41.0)	77 (32.3)	

a: energy intake was calculated from three-day food diaries. CNY: China yuan (1 China yuan = 0.14 US dollar). BMI: body mass index. MET: metabolic equivalent.

**Table 2 nutrients-11-00405-t002:** Factor loadings of foods and food groups in the dominant dietary patterns from food frequency questionnaires administered at 22 weeks gestation and three-day food diaries completed at 24 weeks gestation among 1014 pregnant women in the “Born in Shenyang” cohort.

Dietary Patterns	Food	Factor Loading Coefficient	Variance Explained (%)
**Three-Day Food Diaries**			
Traditional (TFD)	Tubers	0.70	8.0
	Vegetables	0.57	
	Fruits	0.55	
	Rice	0.47	
	Red meat	0.45	
	Eggs	0.26	
	Nuts	0.20	
Sweet foods (TFD)	Pastries and candies	0.63	6.5
	Sweet beverages	0.63	
	Shrimps, crabs and mussels	0.52	
	Fruits	0.31	
	Red meat	0.20	
Fried food–beans (TFD)	Fried foods	0.75	6.0
	Beans and products	0.67	
	Dairy products	0.29	
	Organ meats	−0.23	
Whole grain–seafood (TFD)	Whole grains	0.76	5.9
	Shrimps, crabs and mussels	0.28	
	Nuts	0.24	
	Seaweed	0.20	
	Eggs	−0.27	
	Dairy products	−0.30	
	Rice	−0.37	
Cumulative variance explained (%)			26.4
**Food frequency questionnaire**			
Fish-seafood (FFQ)	Marine fish	0.83	13.2
	Shrimps, crabs and mussels	0.80	
	Freshwater fish	0.77	
	Seaweed	0.66	
	Organ meat	0.51	
Protein–sweets (FFQ)	Dairy products	0.75	10.5
	Milk	0.72	
	Eggs	0.66	
	Beans and products	0.58	
	Nuts	0.40	
	Pastries and candies	0.31	
Traditional (FFQ)	Vegetables	0.86	8.7
	Fruits	0.81	
	Rice	0.49	
	Nuts	0.25	
Cumulative variance explained (%)			32.4

TFD: three-day food diaries; FFQ: food frequency questionnaire.

**Table 3 nutrients-11-00405-t003:** Dietary pattern scores according to maternal characteristics among 1014 pregnant women in the “Born in Shenyang” cohort.

Characteristics	Dietary Pattern Score, Mean (SD)						
	Three-Day Food Diaries				Food Frequency Questionnaire		
	Traditional (TFD)	Sweet Foods (TFD)	Fried Food–Beans (TFD)	Whole Grain–Seafood (TFD)	Fish–Seafood (FFQ)	Protein–Sweets (FFQ)	Traditional (FFQ)
Age at enrollment (years)							
<25	0.51 (1.55)	0.15 (1.46)	0.23 (1.73)	−0.25 (1.14)	−0.62 (1.70)	−0.40 (2.22)	−0.49 (1.71)
25–29	0.06 (1.75)	0.02 (1.41)	−0.01 (1.27)	−0.12 (1.11)	0.12 (3.25)	0.30 (2.69)	0.17 (1.92)
30–34	−0.14 (1.65)	−0.04 (1.17)	0.06 (1.24)	0.15 (1.37)	−0.04 (2.76)	−0.25 (2.10)	−0.17 (1.88)
≥35	−0.04 (1.85)	−0.02 (1.58)	−0.21 (1.14)	0.11 (1.24)	0.00 (2.88)	−0.10 (2.66)	0.11 (1.95)
*p*	0.041	0.786	0.073	0.004	0.333	0.007	0.012
Ethnicity							
Han	−0.02 (1.72)	0.01 (1.37)	0 (1.29)	0.01 (1.22)	0.06 (2.98)	−0.02 (2.41)	−0.02 (1.89)
Minority	0.10 (1.76)	−0.04 (1.31)	0.02 (1.22)	−0.08 (1.32)	−0.31 (2.82)	0.11 (2.80)	0.07 (2.00)
*p*	0.412	0.699	0.831	0.389	0.140	0.540	0.608
Educational attainment							
Middle school	0.17 (1.60)	−0.28 (0.82)	−0.09 (1.24)	−0.14 (1.01)	−0.76 (2.32)	−0.91 (2.23)	−0.68 (2.00)
High school	0.21 (1.78)	−0.07 (1.23)	−0.11 (1.23)	0.02 (0.99)	−0.26 (2.66)	−0.01 (2.49)	−0.19 (2.02)
College	0.00 (1.75)	0.04 (1.35)	0.06 (1.30)	−0.04 (1.26)	0.14 (3.12)	0.08 (2.56)	0.10 (1.89)
Graduate or above	−0.49 (1.46)	0.04 (1.88)	−0.16 (1.21)	0.39 (1.49)	0.06 (2.58)	0.28 (1.83)	0.17 (1.59)
*p*	0.012	0.189	0.210	0.009	0.039	0.003	0.002
Household income per year, CNY							
<30,000	−0.04 (1.77)	−0.15 (1.15)	−0.10 (1.10)	−0.07 (1.07)	0.00 (2.87)	0.05 (2.73)	−0.26 (1.93)
30,000–<50,000	0.15 (1.70)	−0.02 (1.10)	0.10 (1.41)	−0.04 (1.21)	−0.23 (2.52)	−0.21 (2.11)	0.01 (1.81)
50,000–<70,000	−0.01 (1.68)	0.06 (1.40)	0.01 (1.28)	0.05 (1.32)	0.01 (2.49)	0.16 (2.75)	0.03 (1.84)
≥70,000	−0.15 (1.78)	0.06 (1.68)	−0.03 (1.29)	0.12 (1.37)	0.36 (3.91)	0.16 (2.34)	0.35 (1.98)
Missing Value	0.23 (1.44)	0.40 (1.80)	−0.13 (1.31)	−0.32 (1.02)	−0.71 (1.59)	−0.78 (2.00)	−0.69 (2.01)
*p*	0.337	0.119	0.553	0.192	0.121	0.099	0.002
Parity							
0	−0.01 (1.69)	0.03 (1.43)	0.02 (1.28)	0.03 (1.25)	0.02 (2.96)	0.15 (2.56)	0.02 (1.88)
≥1	0.04 (1.83)	−0.12 (1.10)	−0.09 (1.26)	−0.09 (1.20)	−0.08 (2.95)	−0.51 (2.08)	−0.08 (2.00)
*p*	0.702	0.130	0.242	0.201	0.634	<0.001	0.486
Smoking status during pregnancy							
Yes	−0.60 (1.30)	0.09 (1.64)	−0.48 (0.78)	−0.36 (0.74)	0.44 (5.64)	−0.38 (2.51)	−0.36 (1.76)
No	0.01 (1.73)	0.00 (1.36)	0.01 (1.28)	0.01 (1.24)	−0.01 (2.89)	0.01 (2.47)	0.01 (1.91)
*p*	0.136	0.767	0.104	0.211	0.520	0.517	0.416
Pre-pregnancy BMI category, kg/m^2^							
<18.5	−0.12 (1.75)	−0.01 (1.28)	−0.03 (1.30)	−0.25 (0.96)	0.12 (4.17)	0.20 (3.14)	0.45 (1.96)
18.5–<23.0	0.07 (1.72)	−0.01 (1.25)	0.01 (1.31)	0.01 (1.26)	0.06 (2.67)	0.13 (3.14)	0.17 (1.92)
23.0–<25.0	0.04 (1.64)	0.07 (1.81)	−0.02 (1.17)	0.12 (1.43)	−0.02 (3.13)	−0.13 (2.29)	−0.29 (1.82)
≥25.0	−0.15 (1.77)	−0.02 (1.31)	0.00 (1.26)	0.04 (1.17)	−0.22 (2.60)	−0.38 (2.13)	−0.52 (1.76)
*p*	0.363	0.927	0.988	0.076	0.680	0.060	<0.001
Physical Activity, MET-hour/week							
<100	−0.06 (1.71)	0.08 (1.73)	−0.01 (1.44)	0.00 (1.30)	−0.11 (2.93)	−0.28 (2.10)	−0.26 (1.78)
100 to <200	0.02 (1.69)	−0.01 (1.21)	0.01 (1.19)	0.01 (1.19)	−0.08 (2.97)	0.04 (2.51)	0.07 (1.88)
≥200	0.03 (1.83)	−0.09 (1.19)	−0.02 (1.27)	−0.03 (1.27)	0.37 (2.94)	0.27 (2.81)	0.14 (2.11)
*p*	0.783	0.420	0.921	0.926	0.156	0.060	0.035
Energy intake, kcal/day ^a^							
<2100	−0.72 (1.23)	−0.19 (1.15)	−0.23 (1.09)	−0.14 (1.00)	0.00 (2.79)	0.09 (2.59)	0.03 (1.91)
≥2100	1.12 (1.79)	0.30 (1.59)	0.37 (1.45)	0.23 (1.51)	0.00 (3.20)	−0.14 (2.28)	−0.07 (1.91)
*p*	<0.001	<0.001	<0.001	<0.001	0.991	0.145	0.374

a: energy intake was calculated from three-day food diaries. TFD: three-day food diaries; FFQ: food frequency questionnaire; CNY: China yuan (1 China yuan = 0.14 US dollar).

**Table 4 nutrients-11-00405-t004:** Associations of dietary pattern scores, in quartiles, with risk for gestational diabetes mellitus among 1014 women in the “Born in Shenyang” Cohort.

Dietary Patterns	Q1 (*n* = 253)	Q2 (*n* = 253)	Q3 (*n* = 253)	Q4 (*n* = 255)	*p* for Trend
	Reference	Odds Ratio (95% Confidence Interval)			
**Three-Day Food Diaries**					
Traditional (TFD)					
Model 1	1.00	0.69 (0.47, 1.03)	0.85 (0.58, 1.26)	0.42 (0.27, 0.64)	<0.001
Model 2	1.00	0.65 (0.43, 0.98)	0.83 (0.56, 1.23)	0.38 (0.24, 0.60)	<0.001
Model 3	1.00	0.69 (0.45, 1.05)	0.90 (0.58, 1.42)	0.40 (0.23, 0.71)	0.005
Sweet foods (TFD)					
Model 1	1.00	1.04 (0.70, 1.56)	1.03 (0.69, 1.55)	0.76 (0.50, 1.16)	0.231
Model 2	1.00	0.95 (0.62, 1.43)	0.90 (0.59, 1.37)	0.71 (0.46, 1.10)	0.129
Model 3	1.00	0.94 (0.61, 1.44)	0.90 (0.58, 1.39)	0.73 (0.46, 1.16)	0.199
Fried food–beans (TFD)					
Model 1	1.00	1.41 (0.94, 2.12)	1.13 (0.74, 1.71)	1.01 (0.66, 1.54)	0.606
Model 2	1.00	1.41 (0.92, 2.14)	1.18 (0.77, 1.82)	1.07 (0.69, 1.66)	0.871
Model 3	1.00	1.46 (0.95, 2.26)	1.28 (0.82, 2.02)	1.08 (0.67, 1.74)	0.940
Whole grain–seafood (TFD)					
Model 1	1.00	1.27 (0.82, 1.97)	1.68 (1.10, 2.57)	1.75 (1.15, 2.67)	0.006
Model 2	1.00	1.15 (0.73, 1.80)	1.60 (1.03, 2.48)	1.88 (1.22, 2.90)	0.002
Model 3	1.00	1.06 (0.67, 1.68)	1.49 (0.95, 2.35)	1.73 (1.10, 2.74)	0.007
**Food Frequency Questionnaire**					
Fish–seafood (FFQ)					
Model 1	1.00	0.80 (0.53, 1.21)	0.94 (0.62, 1.41)	0.97 (0.65, 1.45)	0.822
Model 2	1.00	0.86 (0.55, 1.35)	1.00 (0.64, 1.59)	0.95 (0.58, 1.57)	0.963
Model 3	1.00	0.87 (0.55, 1.40)	0.99 (0.61, 1.60)	0.95 (0.56, 1.59)	0.986
Protein–sweets (FFQ)					
Model 1	1.00	0.74 (0.49, 1.11)	0.83 (0.55, 1.24)	0.78 (0.52, 1.17)	0.341
Model 2	1.00	0.83 (0.54, 1.28)	1.04 (0.65, 1.66)	1.06 (0.63, 1.78)	0.667
Model 3	1.00	0.94 (0.60, 1.49)	1.23 (0.75, 2.00)	1.18 (0.69, 2.03)	0.434
Traditional (FFQ)					
Model 1	1.00	0.69 (0.47, 1.01)	0.44 (0.29, 0.66)	0.45 (0.30, 0.68)	<0.001
Model 2	1.00	0.68 (0.46, 1.02)	0.43 (0.28, 0.66)	0.43 (0.28, 0.68)	<0.001
Model 3	1.00	0.64 (0.42, 0.98)	0.40 (0.26, 0.64)	0.44 (0.27, 0.70)	<0.001

Model 1: Crude model. Model 2: Adjusted for other dietary patterns derived from the same dietary assessment tool (TFD or FFQ). Model 3: Model 2 plus pre-pregnancy BMI, age, parity, family income, education level, ethnicity, smoking status, total energy intake from three-day food diaries, and physical activity. TFD: three-day food diaries; FFQ: food frequency questionnaire.

**Table 5 nutrients-11-00405-t005:** Associations of dietary patterns score with plasma glucose levels during oral glucose tolerance testing (OGTT).

Dietary Patterns	Blood Glucose Level, β (95% CI)		
	Fasting	1 h after OGTT	2 h after OGTT
**Three-Day Food Diaries**			
Traditional (TFD)			
Model 1	−0.03 (−0.05, −0.01)	−0.11 (−0.18, −0.06)	−0.04 (−0.08, 0.01)
Model 2	−0.03 (−0.05, −0.01)	−0.13 (−0.19, −0.07)	−0.04 (−0.09, 0.00)
Model 3	−0.03 (−0.05, 0.00)	-0.11 (−0.19, −0.03)	−0.02 (−0.08, 0.04)
Whole grain-seafood (TFD)			
Model 1	0.00 (−0.03, 0.03)	0.12 (0.04, 0.21)	0.06 (0.00, 0.13)
Model 2	0.00 (−0.02, 0.03)	0.14 (0.06,0.22)	0.07 (0.01, 0.14)
Model 3	0.00 (−0.03, 0.03)	0.11 (0.02, 0.19)	0.04 (−0.02, 0.11)
**Food Frequency Questionnaire**			
Traditional (FFQ)			
Model 1	−0.03 (−0.05, −0.01)	−0.06 (−0.12, −0.01)	−0.05 (−0.09, −0.01)
Model 2	−0.04 (−0.06, −0.02)	−0.08 (−0.14, −0.01)	−0.06 (−0.11, −0.01)
Model 3	−0.04 (−0.06, −0.02)	−0.07 (−0.13, −0.01)	−0.07 (−0.11, −0.02)

Model 1: Crude model. Model 2: Adjusted for other dietary patterns derived from the same dietary assessment tool (TFD or FFQ). Model 3: Model 2 plus pre-pregnancy BMI, age, parity, family income, education level, ethnicity, smoking status, total energy intake, and physical activity. TFD: three-day food diaries; FFQ: food frequency questionnaire.

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
