# Peer review of "Dietary Patterns during Pregnancy Are Associated with the Risk of Gestational Diabetes Mellitus: Evidence from a Chinese Prospective Birth Cohort Study"

_nutrients, 2019, doi:10.3390/nu11020405_

Round 1
Reviewer 1 Report
The topic is interesting. The adequate attention to diabetes in pregnancy is fundamental for good obtestrics results. The prevalence of diabetes is increasing. Diet is a priority for good control of diabetes and more especially gestational diabetes. The article is well written but a series of questions arise:
-The authors acknowledge that its population is mostly urban. The dietary pattern of the urban population, as well as the lifestyles, is different from the rural one. This can be an important bias since the sample may not be representative of the population. Are the differences between the dietary patterns of the rural and urban population known? Are there substantive differences that can make these results non-extrapopulable?
-The FFQ is validated in its full version, how it can affect having removed numerous items to the validity of the instrument.
-In the current evidence is demonstrating that iron supplements and dietary iron intake are factors associated with the risk of gestational diabetes. The WHO in its 2016 recommendations supports universal iron supplementation. Was this information collected? How would the inclusion of this variable affect the models?
-Did the women know about the rationale of the study? If they knew, the risk of memory or selective recall bias is likely to be very high.
- How many women did not want to respond? What influence can it have on the results? Could they have answered differently than if they had participated?
-How do you know that your sample is representative of the urban population?
Author Response
Response to Reviewer 1 Comments
Point 1: The topic is interesting. The adequate attention to diabetes in pregnancy is fundamental for good obtestrics results. The prevalence of diabetes is increasing. Diet is a priority for good control of diabetes and more especially gestational diabetes.
Response 1: We thank Reviewer 1 for the positive evaluation of our study.
Point 2: The authors acknowledge that its population is mostly urban. The dietary pattern of the urban population, as well as the lifestyles, is different from the rural one. This can be an important bias since the sample may not be representative of the population. Are the differences between the dietary patterns of the rural and urban population known? Are there substantive differences that can make these results non-extrapopulable?
Response 2: Thanks for the comments. According to the National Nutrition and Health Survey in China, dietary patterns in large city, medium- and small-sized cities and rural regions are generally similar in China, thus we think the derived dietary patterns may be representative in both urban and rural areas in southeast of China.Please see Page 12, line 349-350.
Point 3: The FFQ is validated in its full version, how it can affect having removed numerous items to the validity of the instrument.
Response 3: For the FFQ, we combined pork, beef and lamb as a ‘red meat group’, pastry, honey and candy as a ‘pastry and candy group’. According to reviewer’s suggestion, we have validated the FFQ after the combination. The two combined group reported reasonable. Crude Spearman correlation coefficients of consumption frequencies (per day) captured by the two FFQ were from 0.52 (red meat group) to 0.47 (pastry and candy group). We also compared the consumption frequency (per day) between FFQ and TFD for the two food groups, and the Spearman correlation coefficients ranged from 0.52 (red meat group) to 0.43 (pastry and candy group). Other food groups have no food item overlaps with these two groups, thus no changes were observed. We have added these details in the method (page 3, lines 115-116).
Point 4: In the current evidence is demonstrating that iron supplements and dietary iron intake are factors associated with the risk of gestational diabetes. The WHO in its 2016 recommendations supports universal iron supplementation. Was this information collected? How would the inclusion of this variable affect the models?
Response 4: We obtained information of dietary iron intake from three-day food diaries and iron supplements from a single question item respectively. We further adjusted iron supplements and dietary iron intake in our analysis and observed no appreciable changes in the results. Please see Page 4, line 170-171; Page 8, line 245
Point 5: Did the women know about the rationale of the study? If they knew, the risk of memory or selective recall bias is likely to be very high.
Response 5: The women were not told the rationale of the study, and they were asked to report food consumption objectively.
Point 6: How many women did not want to respond? What influence can it have on the results? Could they have answered differently than if they had participated?
Response 6: Among the 2068 women who were eligible to participate in the study, 730 refused to participate. We have compared 1338 women included with 730 women refused to participate. There is no difference between two groups in age, ethnicity and parity. Women who participated in the study had higher educational level, however, the education level was not associated with GDM status in the subjects. Please see Page 4, Line 183-186.
Point 7: How do you know that your sample is representative of the urban population?
Response 7: The 54 hospitals and community health care centers contained all the perinatal care institutes in the urban area of Shenyang. The amount of participants each institute recruited was calculated by using the amount of primary perinatal care of the institute in last year divided by 10. We have added these details into the methods (page 2, paragraph xx, lines 82-85).

Reviewer 2 Report
The MS reports a very interesting regional oriented prospective cohort study on dietary patterns during pregnancy of Chinese patients in northeast China (Shenyang cohort)
1. In this study, the incidence of GDM was 23.5% in the study population. Is it representative for the local population? Or the cohort is enriched in GDM patients.
2. The major question is about smoking habits presented in Table 1. Is 98% of pregnant women smoking? How does it influence dietary habits and the general condition of patients? Do Chinese guidelines involve recommendations of rejecting smoking in pregnancy?
3. The very detailed questionnaire does not involve alcohol consumption.
3. "Fruits and vegetables" is a very general category; can you specify typical content of this group for the studied cohort (specific for northeast China diet). In dietary patterns, tubers are specified, but in FFQ tubers are not selected as an individual category. Is there any reason for this approach.
4. the analysis of calories intake is only for two groups including either < or >2100kcal however a lower GDM risk is associated with higher calories intake. Can you comment on it?
Additionally, physical activity in this study seems to be a factor not preventing GDM, in contrast to general recommendations and some recent studies, including a meta-analysis (Russo LM, et al, Obset Gynecol 2015). Is it connected with relatively low BMI of the study participants?
Author Response
Response to Reviewer 2 Comments
Point 1: The MS reports a very interesting regional oriented prospective cohort study on dietary patterns during pregnancy of Chinese patients in northeast China (Shenyang cohort)
Response 1: We thank Reviewer 2 for the positive evaluation of our study.
Point 2: In this study, the incidence of GDM was 23.5% in the study population. Is it representative for the local population? Or the cohort is enriched in GDM patients.
Response 2: Another large sample birth cohort study in urban Shenyang, the China Medical University Birth Cohort Study, reported a similarly high incidence of GDM (21.7%). Thus we believe that the incidence may be representative for the local population of the urban area in Shenyang.
Point 3: The major question is about smoking habits presented in Table 1. Is 98% of pregnant women smoking? How does it influence dietary habits and the general condition of patients? Do Chinese guidelines involve recommendations of rejecting smoking in pregnancy?
Response 3: We apologised for the mistake; 98.2% of pregnant women did not smoke during pregnancy and only 1.8% of pregnant women smoked. We have corrected it in the manuscript.
Point 4: The very detailed questionnaire does not involve alcohol consumption.
Response 4: The FFQ and the three-day food diaries did not have questions on alcohol consumption. However, we investigated women’s alcohol consumption history during pregnancy using single item :‘Have you ever drink alcohol during pregnancy’
and few women reported to have consumed during pregnancy.
Point 5: "Fruits and vegetables" is a very general category; can you specify typical content of this group for the studied cohort (specific for northeast China diet). In dietary patterns, tubers are specified, but in FFQ tubers are not selected as an individual category. Is there any reason for this approach.
Response 5: We have listed content of each food group in Table S1 of the supplementary file. Both in the FFQ and three-day food diaries, tubers are selected as an individual category. However, the derived tradition dietary pattern (FFQ) did not involve in tubers.
Point 6: the analysis of calories intake is only for two groups including either < or >2100kcal however a lower GDM risk is associated with higher calories intake. Can you comment on it?
Response 6: Thank you for the suggestion. One of the possible reasons could be that in our study subjects, high energy intake (≥2100 kcal/d) was also associated lower age which is a protective factor for GDM. However, further studies are needed to investigate this. We have added these statements in the discussion, please see page 11, line 295-298.
Point 7: Additionally, physical activity in this study seems to be a factor not preventing GDM, in contrast to general recommendations and some recent studies, including a meta-analysis (Russo LM, et al, Obset Gynecol 2015). Is it connected with relatively low BMI of the study participants?
Response 7: In our study, physical activity level was neither associated with GDM nor associated with gestational weight gain. The possible reason could be that the Pregnancy Physical Activity Questionnaire (PPAQ) measured physical activity level of in a short time(7 days) before OGTT, thus may not represent the physical activity level during the whole early pregnancy. We have added these statements in the discussion, please see page 11, line 298-301.
